# Identification of Atmospheric Transport and Dispersion of Asian Dust Storms

Raegyung Ha[1], Amarjargal Baatar[2], Yongjae Yu[1,2]

[1]Department of Astronomy, Space Science, and Geology, Chungnam National University, Daejeon, 34134, Korea
[2]Department of Geology & Environmental Sciences, Chungnam National University, Daejeon, 34134, Korea

*Correspondence to*: Yongjae Yu (youngjaeyu@cnu.ac.kr)

**Abstract.** Backward trajectories of individual Asian dust storm (ADS) events were calculated using the hybrid single particle Lagrangian integrated trajectory (HYSPLIT) at four representative stations in Korea. A total of 743 ADS events and associated 2229 (endings of altitudes at 1000 m, 1500 m, and 2000 m per ADS event) backward trajectories from four stations were traced from January 2003 to August 2015. Regardless of the locations of the observed stations and the threshold time divide, recent increase of ADS occurrence rate was statistically significant in 99.9 % confidence limit. Winters and springs were high occurrence season for the ADS, while the ADS rarely occurred in summers. Angular distributions of dust transport indicated a dominance of northwesterly, as more than two-thirds of ADS events are azimuthally confined from 290°−340°. In addition, there is a tendency for stronger PM10 dust air concentration to be from the northwest. We found a strong inverse correlation between number of days with ADS events and cumulative PM10 dust air concentration, indicating that the total amount of cumulative PM10 discharge was rather constant over time. If so, relatively shorter transport distance and more continental dust passage through Shandong peninsular would yield less but stronger PM10 concentration in shorter transport path.

## 1 Introduction

Asian dust storm (ADS) originates mostly in the Gobi and Taklamakan deserts, where high speed surface winds and intense dust storms soared dry, dense clouds of fine grained surface material. Once elevated, ADS spreads in the stratosphere, often reaches to the upper troposphere to form the dust devil (Yumimoto et al., 2009). These dusty clouds are then carried eastwards by prevailing westerly winds and pass over from China, Korea, Taiwan, Japan, and to the North Pacific in order (Hsu et al., 2012). These dusty clouds were eventually transported more than one full circuit around the globe in about two weeks (Uno et al., 2009). Such long range transport is possible as the dusty clouds are transported in atmospheric dusty layers (Tratt et al., 2001; Schepanski et al., 2009; Uno et al., 2009). It has been estimated that ADS contributes about 4 % of the global dust emission (Mahowald et al., 2010).

ADS contained surficial minerals of natural origin (e.g., weathered soils) as well as pollutants of anthropogenic origin such as black carbon, heavy metals, and sulfates (Guo et al., 2004; Hsu et al., 2004; Ramana et al. 2010). These pollutants have impacts on climate variations as black carbons absorb solar radiation (Jacobson 2001, 2012) whilst sulfates scatter solar radiation

energy (Ramana et al. 2010). Deposition of anthropogenic pollutants also affects the human epidermal keratinocytes (Choi et al., 2011), dispersion of bacterial cells (Yamaguchi et al., 2012), and marine biogeochemical environment (Lin et al. 2007; Hsu et al. 2009). In addition, long range transport of ADS influences cardiovascular disease in Taiwan (Chen and Yang, 2005), cardiopulmonary emergence rate in Taiwan (Chan et al., 2008), worsening Asthma in Japan (Watanabe et al., 2011), daily

mortality in Korea (Lee et al., 2013), and food toxicity in Japan (Kobayashi et al., 2015). Hence it is natural to raise public concern on the possible adverse effects of ADS in East Asia.

ADS influences air quality and the local ecological environment including vegetation and soil. It accelerated the process of desertification by increasing the rate of evaporation. Most of all, it has been observed that the frequency of ADS occurrence in East Asia increased as a result of desertification, over grazing and over farming, irrigation, and lack of precipitation in

central Asia. Of course, it is also true that ADS is dependent on the synoptic climatic conditions such as cyclone activity, air temperature, and precipitation in source area (Natsagdorj et al. 2003; Qian et al. 2004; Hsu et al., 2013).

Over the past few decades, variation of ADS and its climate control has been explored (Natsagdorj et al. 2003; Qian et al. 2004; Kim et al. 2008; Lee et al., 2010; Zhao et al., 2010). For instance, Natsagdorj et al. (2003) compiled dust storms in Mongolia from 1937 to 1999, on the basis of observational data from 49 stations, found that annual mean number of days with dust

storms were 20−37 days in the Gobi deserts and nearby arid area. Wang et al. (2005) defined three different types of dust storms and their characteristics in China, using data from 701 meteorological observation stations from 1954 to 2000. It should be highlighted that the number of ADS events increased by more than twice from 1960 to 2000 (Zhang et al. 2003; Wang et al. 2007, 2008).

Previous studies in Korea focused on the characteristic of dust particles and magnetic concentration of dust particles in ADS

(Chun et al. 2001, 2008; Chung et al., 2003; Lee et al. 2006; Kim et al. 2008; Lee et al., 2013). In the present study, we trace the air parcel trajectories of ADS using the hybrid single particle Lagrangian integrated trajectory (HYSPLIT) model (Draxler and Hess, 1998). Temporal and spatial variation of ADS allows a more comprehensive view of the dust generation and transport in East Asia. In particular, tracing the ADS trajectories in Korea is pivotal because Korea is the first encountered out of source region of ADS on its westerly dominating dust transport. It is the purpose of this study to gain insight into how atmospheric

transport, dispersion, and deposition of dusty particulates occurred in Eastern Asia.

## 2 The HYSPLIT Model

The HYSPLIT has evolved from the earliest model in 1982 (Draxler and Taylor, 1982) from modelling long-range air parcel trajectories into simulations of pollutant transportation, dispersion, and deposition over global scales. The spirit of Lagrangian HYSPLIT model relies on the determination of air concentration, as a cumulative summation of dust flow per unit grid cell.

Each dust flow is considered as an independent particle flow puffed by advection, and is represented by its trajectory. Backward trajectories are constructed on the basis of Stochasitc Time Inverted Lagrangian Transport (STILT) model. The STILT is a

widely used model for tracing atmospheric mixing between the source and the receptor point in terms of 2-dimensional upstream surficial fluxes.

As an open source, the HYSPLIT is available on the Web through the ARL READY system (http://ready.arl.noaa.gov/HYSPLIT.php), operated by the National Oceanic and Atmospheric Administration (NOAA) Air Resource Laboratory (ARL). In the present study, we used HYSPLIT from the most recent model in September 2015 (Stein et al., 2015).

The HYSPLIT model requires the meteorological data and vertical movement of atmospheric circulation as input, and it displays the analysis of the simulation outputs (Stein et al., 2015). The HYSPLIT model describes transport and dispersion dynamics of aerosol, incorporating boundary stability determined by turbulent velocity, wind-blown dust emission algorithm, convectional plume rise produced by buoyancy of heat, wind velocity, atmospheric friction velocity, and in-cloud wet scavenging. One great advantage of using the Lagrangian HYSPLIT is that both forward and backward trajectories are available with local or global airflow patterns to interpret the transport of pollutants. The HYSPLIT model is continuously evolving to cope with turbulent mixing process and to incorporate higher temporal frequency data available from the meteorological data (Stein et al., 2015).

## 3 Data and Analysis

Korea Meteorological Administration (KMA) operates 28 local stations where each station records meteorological data including air pollution monitoring (http://web.kma.go.kr/eng/). In particular, KMA posted online real time insitu dust air concentration measurements in major stations. Among 28 stations, we compiled data collected from four representative stations including Baekryeongdo (BR, 37°58'00" N, 124°38'00" E), Kosan (KS, 33°17'00" N, 126°09'00" E), Ulreungdo (UR, 37°03'00" N, 130°55'00" E), and Daejeon (DJ, 36°22'8" N, 127°20'49" E) (Fig. 1a). These stations are located at the western front (BR), southern edge (KS), eastern tail (UR), and central region (DJ) in South Korea, respectively. They were selected on the basis of spatial distribution and longer decadal coverage of observation periods.

In general, PM10 dust air concentration less than 25 µg m$^{-3}$ is considered to be recommended by the World Health Organization. According to KMA, advisory warning is issued when the hourly averaged PM10 dust air concentration is expected to exceed 150 µg m$^{-3}$ for an hour. When the hourly averaged PM10 dust air concentration is expected to exceed 400 µg m$^{-3}$ for an hour, more significant danger warning is issued. In the present study, individual ADS event was accounted as the day with advisory warning.

To trace the ADS provenance source, online version of backward trajectories HYSPLIT model (September 2015) was used. We used inputs of meteorological data from the National Centers for Environmental Prediction (NCEP; http://www.ncep.noaa.gov/) and the National Center for Atmospheric Research (NCAR: https://ncar.ucar.edu/) Reanalysis-1. Vertical motion of aerosol was adopted from the "Model Vertical Velocity" option on the HYSPLIT model (September 2015). Results of backward trajectories were displayed on ArcGIS program. The NCEP/NCAR Reanalysis-1

 2.5° every 6 hours, regarding the vertical distribution of global aerosol and cloud from 1958 to present. Air mass trajectory analysis was carried out using the HYSPLIT model with global data assimilation system (Stein et al., 2015), at the time of ADS events (Fig. 1b). Airborne pollutants are included in clouds as dusty clumps along with water vapour droplets. In other

words, emission, transport and dispersion of airborne pollutants concentrate in common cloud heights of 1-2 km. Hence, the trajectories of air transport at altitudes of 1000, 1500 and 2000 m were traced for 72 hours (Fig. 1b).

## 4 Result

Backward trajectories of individual ADS events were calculated using the HYSPLIT at four representative stations at BR, DJ,

KS, and UR (Fig. 1a). The data archive includes a collection of HYPLIT model since 2003 based on NCEP/NCAR reanalysis 1 at the time of ADS events. Trajectories of dust transport at altitudes of 1000 m, 1500 m, and 2000 m were traced for 72 hours (Fig. 1b). A total of 743 ADS events and associated 2229 (three different endings of altitude per ADS event) backward trajectories from four stations were traced from January 2003 to August 2015 (Table 1). Then, ADS was classified into six transport path (Fig. 1b) on the basis of wind azimuth during the first 24 hours of each ADS event as N−NE (northerly 0° to

northeasterly 45), N−NW (northerly 360° to northwesterly 315°), W−NW (westerly 270° to northwesterly 315°), W−SW (westerly 270° to southwesterly 235°), S−SW (southerly 180° to southwesterly 235°), and S−SE (southerly 180° to southeasterly 135°) (Table 1).

Annual cycles of number of days each year with ADS events ($n_{ADS}$) are displayed (Fig. 2). A modern data set with PM10 observation from 2003 to the present was combined with an old dataset without PM10 (Fig. 2). For the past 13 years,

cumulative number of days with ADS events was 111 at BR, 220 at DJ, 257 at KS, and 155 at UR (Fig. 3a, Table 2). For each station, annual and monthly variations of ($n_{ADS}$) were similar with one another (Fig. 3b, c). However, it is interesting that ($n_{ADS}$) of BR was systematically lower than that for other stations (Fig. 3b). For the past 13 years, the lowest frequency of ADS occurred in 2012 (Fig. 3b). Winters and springs were high occurrence season for the ADS, while the ADS rarely occurred in summers (Fig. 3c).

Temporal variations of dust air concentration observed in each station were represented with time (Fig. 4 and Fig. 5). For each station, distribution of individual PM10 dust air concentration was plotted as a function of time in lower panel (Fig. 4 and Fig. 5). In upper panels (Fig. 4 and Fig. 5), results were rearranged in boxplots where central box represents the interquartile of annual mean PM10 dust air concentration and whisker lines are extending beyond the maximum and the minimum. For BR, a total of 111 ADS events from 2003 to 2015 showed mean PM10 dust air concentration of 424.7±341.1 µg m$^{-3}$ (Fig. 4a, Table

2). On the other hand, values of mean PM10 dust air concentration in other stations were smaller than those at BR (Table 2). For instance, DR, KS, and UR recorded mean PM10 dust air concentration of 189.5±94.2 µg m$^{-3}$ (Fig. 4b), 190.7±62.3 µg m$^{-}$

$^3$ (Fig. 4c), and 188.3±55.7 µg m$^{-3}$ (Fig. 4d), respectively. Danger warning issued days with PM10 dust air concentration exceeding 400 µg m$^{-3}$were 37 for BR (33.3 %), 5 for DJ (2.3 %), 7 for KS (2.7 %), and 3 for UR (1.9 %). Values of PM10 dust air concentration were tend to be larger over the spring seasons (Fig. 5).

On a log−log plot, PM10 dust air concentration observed from BR plot in a line with a slope of -1.5, meaning that an order increase in PM10 dust air concentration correlates with a 50-fold decrease in ADS occurrence (Fig. 6a). For UR, the slope of maximum asymptotic power fitting yielded -4.0, an order increase in PM10 dust air concentration correlates with a four order decrease in ADS occurrence (Fig. 6a). Results for DJ and KS were confined within the trends of BR and UR (Fig. 6a). They were definitely not linear, but are tailed towards stronger values of PM10 dust air concentration for lower occurrence (Fig. 6a). We have constructed cumulative probability distribution functions for each station (Fig. 6b). Values of median PM10 dust air concentration were 318.72 µg m$^{-3}$ for BR, 162.13 µg m$^{-3}$ for DJ, 164.49 µg m$^{-3}$ for KS, and 171.62 µg m$^{-3}$ for UR (Fig. 6b). Tracing the dust transport using the HYSPLIT was represented with angle histogram plot, which is a polar plot showing the distribution of prevailing wind in angle bins of 5 degrees (Fig. 7). For the past 13 years, spatial distribution of ADS events is prominently northwesterly (Fig. 7).

**5 Discussion**

Because of its shortest operation history, only the modern data are available in BR (Fig. 2a). More extended temporal coverages over 50 years of ADS events were available in DJ (Fig. 2b), KS (Fig. 2c), and UR (Fig. 2d). It is eye catching that the annual occurrence rate of ADS events has been increased recently (Fig. 2). For instance, annual mean occurrence rate of ADS in Daejeon was 4.1±4.0 from 1960 to 2000 and 17.2±6.5 from 2001 to 2015, which were significantly different (p = 7.611±10$^{-7}$) each other according to the criteria of Welch's t−test with a significance of 99.9 % (Table 3). Regardless of the locations of the observed stations and the threshold time divide (either 1998 or 2001), recent increase of ADS events was statistically significant in 99.9 % confidence limit. It is also apparent that there may be also a decline in ADS occurrence with respect to 2007/2008. Results for Welch's t−test are in the order of 10$^{-2}$, slightly over the statistical threshold for the acceptance of the null hypothesis. Despite its failure of the null hypothesis, we cannot completely rule out the possibility as the temporal separation in 2007/2008 leaves only 8 data-points for recent intervals (Table 3).

Year to year variations (Fig. 3b) and month to month variations (Fig. 3c) of ADS events were compared. It is apparent that ADS is heavily concentrated in dry seasons, from February to June (Fig. 3c). The highest dust air concentration occurred was 2371 µg m$^{-3}$ on April 8, 2006 at BR, 862 µg m$^{-3}$ on September 12, 2010 at DJ, 1342 µg m$^{-3}$ on April 2, 2007 at KS, and 440 µg m$^{-3}$ on April 02, 2007 at UR (Fig. 4 and Fig. 5). Occurrence of more frequent ADS and its seasonality was also documented in neighboring countries including Japan and Taiwan (Yang, 2002; Watanabe et al, 2011; Chien et al., 2012; Kimura, 2012). Then, it can be understood that seasonal variation of ADS is regional, natural phenomenon in Eastern part of Asia.

Several factors are known to be closely related with the occurrence and strength of dust storms including extinction of pastureland, abandoning cropland without vegetation cover, overexploitation of forests and shrubs, enhancement of mining activities, unregulated wild roads desertification, increase of human or factory transport, and decline of annual precipitation (Xuan et al., 2004; Aoki et al., 2005; Bian et al., 2011). Increase of desertification results in high dust emission from the places

with less vegetation cover with dry and loose sandy soils in the Gobi and Taklamakan deserts (Laurent et al., 2005, 2006; Zhang et al., 2008). Recent increase of ADS events is contemporaneous to the increased desertification in the Gobi and Taklamakan deserts. For instance, numbers of the dusty days in the Gobi deserts have been almost tripled in recent years when compared to those in 1960's (Natsagdorj et al., 2003). In another occasion, it has been reported that sandy desertification in North China increased rapidly with mean annual areal expansion of 2460 km$^2$ (Zhang et al., 2008). In addition, decrease on

the amount of soil moisture and increase of mean wind speed provide more frequent generations of dust storms (Natsagdorj et al. 2003). It should be highlighted that the highest frequency of ADS arose from the Gobi deserts, mostly in spring season due to the development of lower air temperature in winter season and high frequencies of cyclone activities (Qian et al., 2002). In addition to natural pedogenic enhancement, we cannot ignore the contribution of anthropogenic particulate matters supplied by fossil fuel combustion, coal burning and industrial plants. Although anthropogenic particulate matters represent only 5-30%

of ADS volumetrically, they are harmful as they have a strong tendency to react with heavy metals preferentially. Considering on-going demand for the fossil-fuel combustion, it is reasonable to suggest that pollutants of anthropogenic origin are also responsible for the increase of ADS.

To evaluate the differences between the data sets observed from four different stations, a Kolmogorov−Smirnov test of a non-parametric and distribution free statistics was applied. When the value of P is insignificant (e.g., P < 0.01), we can reject the

20 null hypothesis of no difference between the two data sets. According to the Kolmogorov−Smirnov test, data sets KS and UR are similar (P=0.7040) and those between DJ and KS are somewhat similar (P=0.0280) in 98 % confidence limit. Contrary to the initial impression based on the correlation between ADS occurrence and PM10 dust air concentration (Fig. 6), however, all other pairs of data sets are statistically different (Table 4). Angular distribution of dust transport indicates a dominance of northwesterly (Fig. 7). In fact, the prevailing ADS is from a narrow sector between 290° and 340° (Fig. 7). There is a tendency

for stronger dust densities to be from the northwest (Fig. 7).

The HYSPLIT backward trajectories at different altitudes of 1000 m, 1500 m, and 2000 m were counted as individual path in the present study. They do not show a meaningful difference statistically, implying that atmospheric turbulent mixing was minimal. Such directional consistency for different altitudes of the HYSPLIT model might result from the relatively low and flat geographic conditions. For instance, both eastern China and western Korea are low in elevation, with bridging shallow

Yellow Sea which extends 900 km in North-South directions and 700 km in East-West directions.

As ADS is a randomly occurring natural hazard, populations of individual ADS events ($n_{ADS}$) should follow the exponential probability density distribution with the weaker dust air concentration occurs more frequently (Fig. 6a). As anticipated, ADS

occurrence and PM10 dust air concentration displayed an inverse power relation in all four stations (Fig. 6a). But why did each station record a different power relation?

Thermodynamic equilibrium of wind−blown dust requires competing balance among uprising buoyancy, gravitational settlement, and flow resistant drag force. Strong inverse correlation between the number of days with ADS events ($n_{ADS}$) and cumulative PM10 dust air concentration implies that the total amount of cumulative PM10 discharge is rather constant over time. As a result of the nearly constant PM10 discharge, relatively shorter transport distance along with more continental dust passage through Shandong peninsular would yield less but stronger PM10 dust air concentration in BR (Fig. 8a). In other words, longer settlement time intervals were required for DJ, KS, and UR (Fig. 8a), simply because they are located farther than BR from the dust sources (Fig. 1).

Mean PM10 dust air concentration was estimated by dividing the total amount of cumulative PM10 flux to the total number of ADS events ($n_{ADS}$). In fact, mean PM10 dust air concentration ($PM10_{mean}$) is equivalent to the arithmetic mean of PM10 dust air concentration for each station. However, it is true that individual dust air concentration distribution is far from being Gaussian or Log−Normal (Fig. 6b). Instead, median PM10 dust air concentration ($PM10_{median}$) estimated from the cumulative dust air concentration distribution is useful to reflect individual PM10 dust air concentration distribution. Regardless of distribution of natural hazards, the median is inherently more stable than the mean with respect to the uncertainty and will change less over time (e.g., Atkinson and Goda, 2011). Nonetheless, as far as the present study is concerned, $PM10_{mean}$ and $PM10_{median}$ can both reflect the mean PM10 dust air concentration as they are positively correlated (Fig. 8b).

## 6 Conclusions

The present study dealt with the Asian dust storms (ADS) outbreaks affecting Korea from January 2003 to August 2015. A total of 743 ADS air parcel backward trajectories reaching to Korea were identified by means of Lagrangian integrated trajectory (HYSPLIT) at three different ending altitudes at 1000, 1500, and 2000 m. In all four stations where ADS was monitored, we found that ADS occurrence rate was increased recently. Such increase of ADS occurrence was statistically significant in 99.9 % confidence level regardless of the threshold time divide of 1997/98 or 2000/01. Monthly variation of ADS occurrence was definitely non-uniform, as ADS was mostly concentrated in colder seasons of winters and springs. Instead, ADS events rarely occurred from June to September. Majorities of ADS events are azimuthally confined in narrow intervals of 290−340° on angle histograms, indicating that northwesterly distribution of dust transport was prominent. Such angular dependence of ADS occurrence agrees well with the higher PM10 dust air concentration from the northwest. We propose that the total amount of cumulative PM10 discharge was rather constant over time in Korea, as there is an inverse correlation between ADS occurrence and PM10 dust air concentration. Such constant PM10 flux allows weaker PM10 concentration for longer transport, and vice versa.

*Author Contribution.* RH, AB, and YY carried out data compilation, statistical analysis and drafted the manuscript. RH and YY conceived of the study and participated in its design and coordination. RH and YY was in charge of the climatological interpretation. All authors read and approved the final manuscript.

*Acknowledgements.* This work was supported by This work was supported by the Polar Academic Program (PD1601), Korea Polar Research Institute, 2016. We thanks to Doohee Jeong, Hanul Kim, and Hoabin Hong for providing technical assistance in using scanning electron microscopy.

*Competing Interests.* The author declares that they have no conflict of interest.

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

**Table 1: Azimuthal dependence of ADS transport path**

|  | BR | DJ | KS | UR | Total |
|---|---|---|---|---|---|
| N−NW (northerly 360° to northwesterly 315°) | 184 | 239 | 386 | 172 | 981 |
| W−NW (westerly 270° to northwesterly 315°) | 113 | 332 | 320 | 193 | 958 |
| W−SW (westerly 270° to southwesterly 235°) | 14 | 46 | 31 | 50 | 141 |
| S−SW (southerly 180° to southwesterly 235°) | 7 | 13 | 10 | 24 | 54 |
| N−NE (northerly 0° to easterly 45°) | 9 | 17 | 16 | 10 | 52 |
| S−SE (southerly 180° to southeasterly 135°) | 3 | 5 | 3 | 12 | 23 |
| Unclassified (45° to 135°) | 3 | 8 | 5 | 4 | 20 |
| Total | 333 | 660 | 771 | 465 | 2229 |

For each ADS event, backward trajectories for three different endings of altitude were traced from January 2003 to August 2015.

**Table 2: Summary of ADS discharge from January 2003 to August 2015**

|  | BR | DJ | KS | UR |
|---|---|---|---|---|
| number of days ($n_{ADS}$) with advisory warning issued ADS events (PM10 dust air concentration $\geq 150$ μg m$^{-3}$) | 111 | 220 | 257 | 155 |
| number of days ($n_{ADS}$) with danger warning issued ADS events (PM10 dust air concentration $\geq 400$ μg m$^{-3}$) | 37 | 5 | 7 | 3 |
| cumulative sum of PM10 dust air concentration in (μg m$^{-3}$) | 47,147 | 34,123 | 36,021 | 29,383 |
| annual mean of cumulative sum of PM10 dust air concentration in (μg m$^{-3}$) | 3626.7 | 2624.8 | 2770.8 | 2260.2 |
| mean PM10 dust air concentration (PM10 $_{mean}$) of ADS in (μg m$^{-3}$) | 424.75 | 155.10 | 140.16 | 189.57 |
| median PM10 dust air concentration (PM10 $_{median}$) of ADS in (μg m$^{-3}$) | 318.72 | 162.13 | 164.49 | 171.62 |

**Table 3: Statistical significance of the increased occurrence rate of ADS events.**

| Location | Interval | Years | Mean | 1 σ | Welch *t*–test | *P* |
|---|---|---|---|---|---|---|
| DJ | 1960–2007 | 48 | 6.563 | 7.158 | 0.701 | $2.393\times10^{-2}$ |
| | 2008–2015 | 8 | 13.750 | 7.005 | | |
| | 1960–2000 | 41 | 4.073 | 3.965 | 2.842 | $7.611\times10^{-7}$ |
| | 2001–2015 | 15 | 17.200 | 6.461 | | |
| | 1960–1997 | 36 | 3.833 | 3.753 | 2.627 | $2.930\times10^{-7}$ |
| | 1998–2015 | 18 | 15.889 | 6.703 | | |
| UR | 1962–2007 | 46 | 4.478 | 7.996 | 0.457 | $8.487\times10^{-2}$ |
| | 2008–2015 | 8 | 9.500 | 6.740 | | |
| | 1962–2000 | 39 | 1.744 | 2.302 | 2.293 | $1.389\times10^{-4}$ |
| | 2001–2015 | 15 | 14.267 | 9.392 | | |
| | 1962–1997 | 36 | 1.444 | 2.076 | 2.110 | $6.816\times10^{-5}$ |
| | 1998–2015 | 18 | 12.778 | 9.214 | | |
| KS | 1988–2007 | 20 | 10.550 | 11.821 | 0.448 | $3.049\times10^{-2}$ |
| | 2008–2015 | 8 | 19.750 | 8.225 | | |
| | 1988–2000 | 13 | 4.308 | 3.794 | 1.431 | $1.987\times10^{-5}$ |
| | 2001–2015 | 15 | 20.867 | 10.453 | | |
| | 1988–1997 | 10 | 3.200 | 2.658 | 1.202 | $1.115\times10^{-5}$ |
| | 1998–2015 | 18 | 18.722 | 10.845 | | |
| BR | 2001–2007 | 7 | 14.143 | 7.925 | 0.610 | $7.047\times10^{-2}$ |
| | 2008–2015 | 8 | 7.250 | 4.301 | | |

The Welch's t-test is used to test the hypothesis that two populations have equal means. The Welch's t-test is an extension of Student's t-test and is more reliable when the two sample sets have unequal variances and unequal sample sizes. The value of
5  "p" is the probability of obtaining significantly different sample means between two data sets.

**Table 4: Kolmogorov-Smirnov comparison of two data sets.**

|     | BR | DJ | KS | UR |
|-----|-----|-----|-----|-----|
| BR  | D=0.0000 | D=0.6787 | D=0.6316 | D=0.5954 |
|     | P=1.0000 | P=0.0000 | P=0.0000 | P=0.0000 |
| DJ  |     | D=0.0000 | D=0.1326 | D=0.1789 |
|     |     | P=1.0000 | P=0.0280 | P=0.0050 |
| KS  |     |     | D=0.0000 | D=0.0707 |
|     |     |     | P=1.0000 | P=0.7040 |
| UR  |     |     |     | D=0.0000 |
|     |     |     |     | P=1.0000 |

The maximum difference between the cumulative distribution (D) and corresponding probability (P) according to the Kolmogorov−Smirnov test.

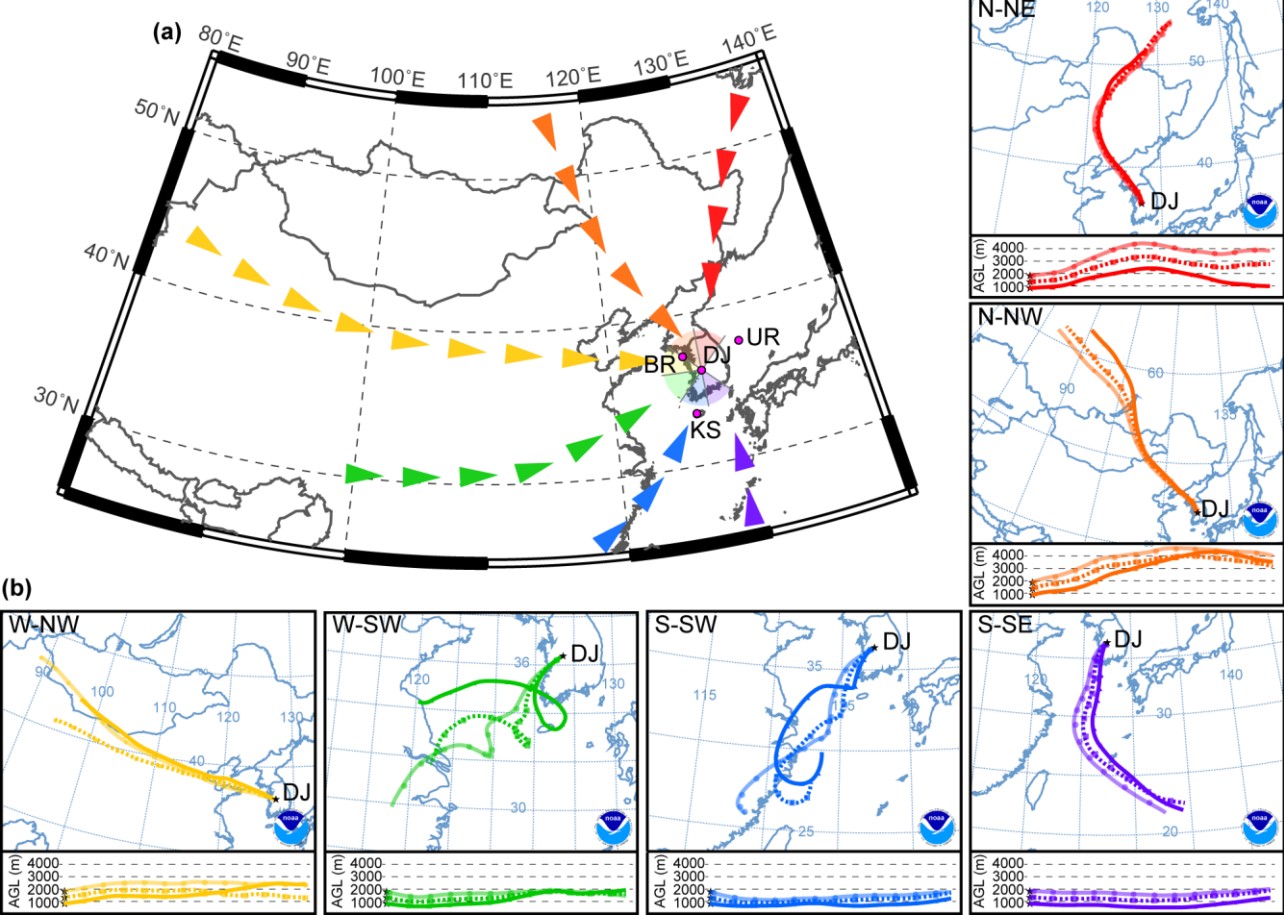

**Figure 1: Figure 1: (a)** Schematic diagram of dust transport path, BR: Baekryeongdo, DJ: Daejeon, KS: Kosan, UR: Ulreongdo. **(b)** Representative examples of HYSPLIT backward trajectories ending at DJ. Trajectories of dust transport at 1000 m, 1500 m, and 2000 m above mean sea level (AMSL) were traced for 72 hours. ADS was classified into six transport path on the basis of wind azimuth during the first 24 hours of each ADS event as N−NE (northerly 0° to northeasterly 45°), N−NW (northerly 360° to northwesterly 315°), W−NW (westerly 270° to northwesterly 315°), W−SW (westerly 270° to southwesterly 235°), S−SW (southerly 180° to southwesterly 235°), and S−SE (southerly 180° to southeasterly 135°).

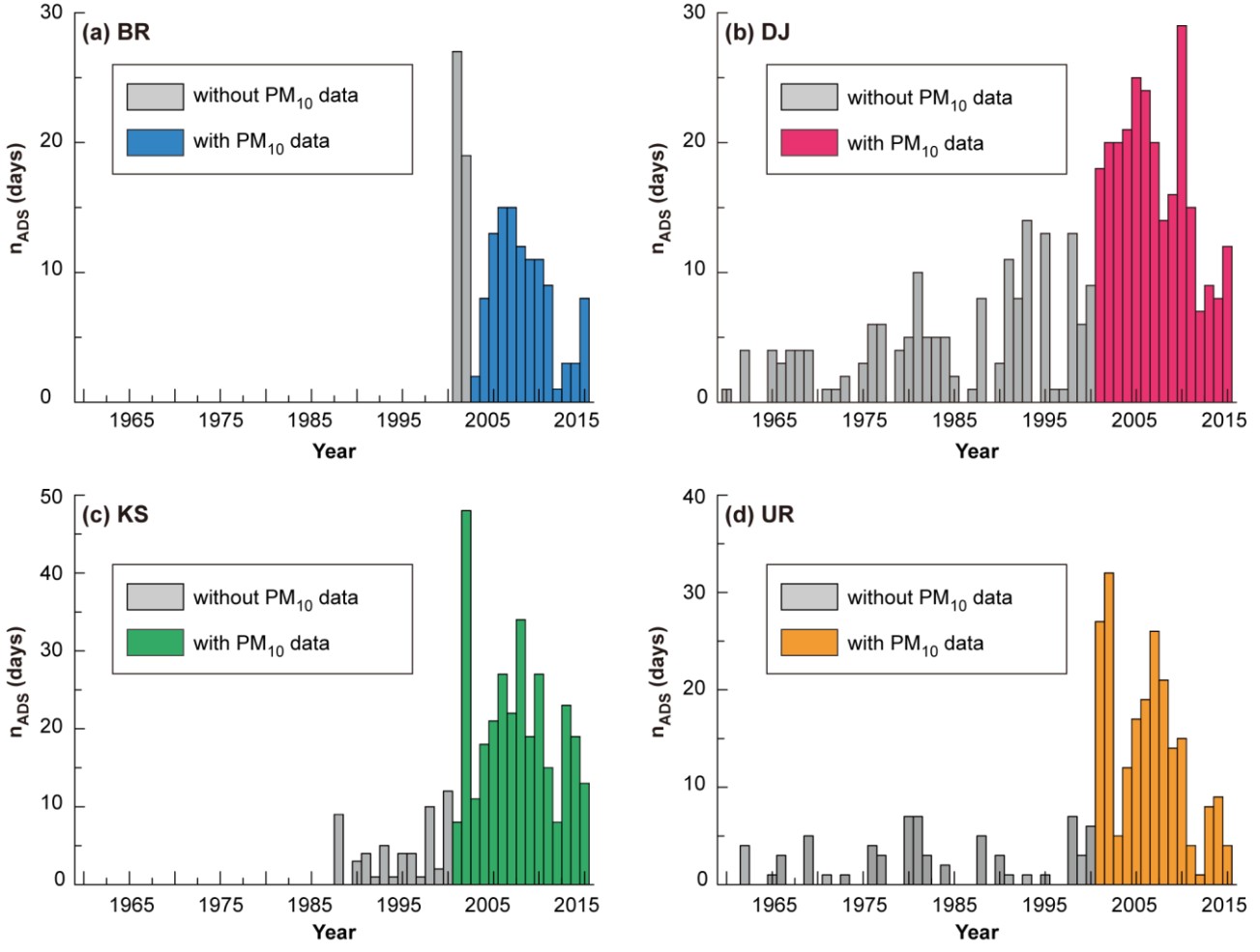

Figure 2: Number of days each year with ADS events (n$_{ADS}$) at (a) BR, (b) DJ, (c) KS, (d) UR. Modern data set includes the PM10 observations.

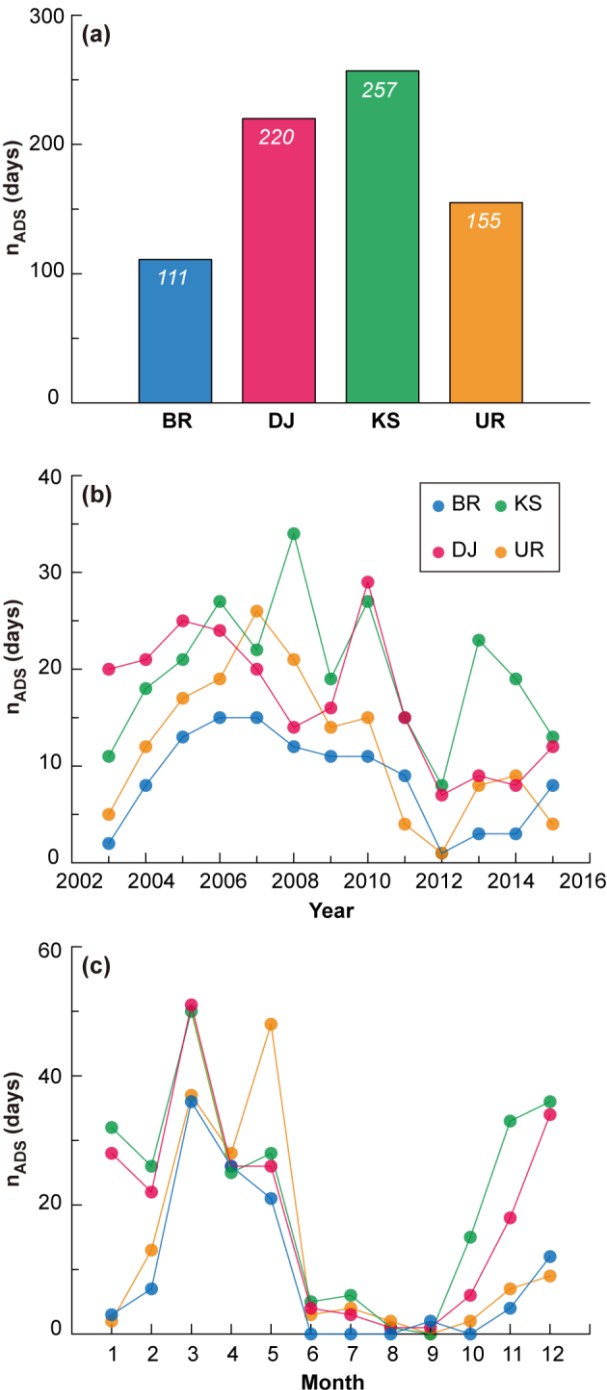

**Figure 3: Comparison of ADS events for the past 13 years. (a) number of days each year with ADS events (n$_{ADS}$) in histogram, (b) annual variations of ADS events, (c) monthly variations of ADS events.**

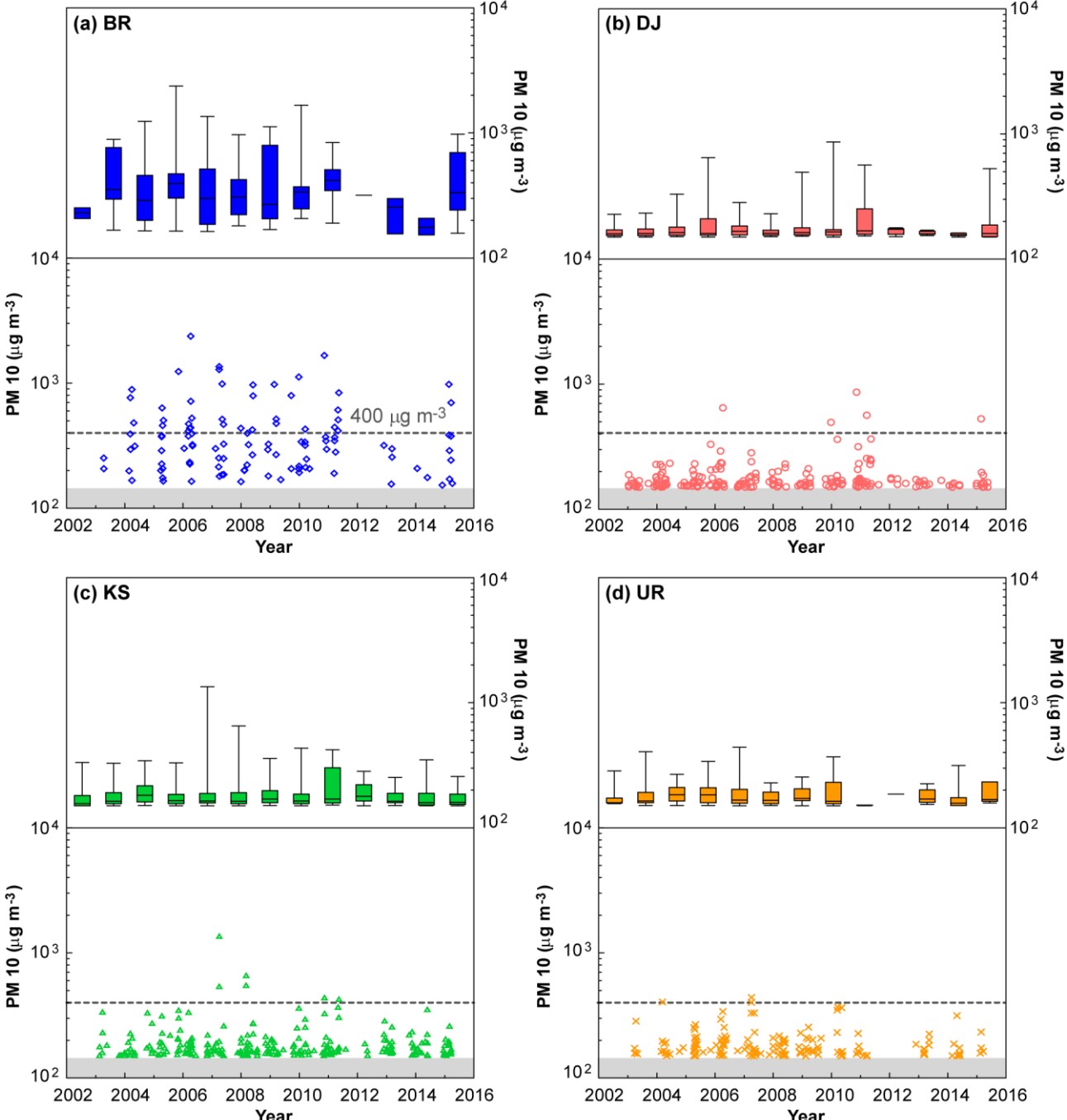

**Figure 4: Annual variations of PM10 dust air concentration at (a) BR, (b) DJ, (c) KS, (d) UR. The lower panel shows a distribution of individual dust** air concentration**. The upper panel displays boxplots where central box represents the inter-quartile of annual mean dust** air concentration **and whisker lines are extending beyond the maximum and the minimum. In the present study, individual ADS event was accounted as the day with advisory warning with PM10 dust air concentration exceeds 150 µg m⁻³ for an hour. Another reference line for more significant danger warning with PM10 dust air concentration exceeds 400 µg m⁻³ for an hour was marked for comparison.**

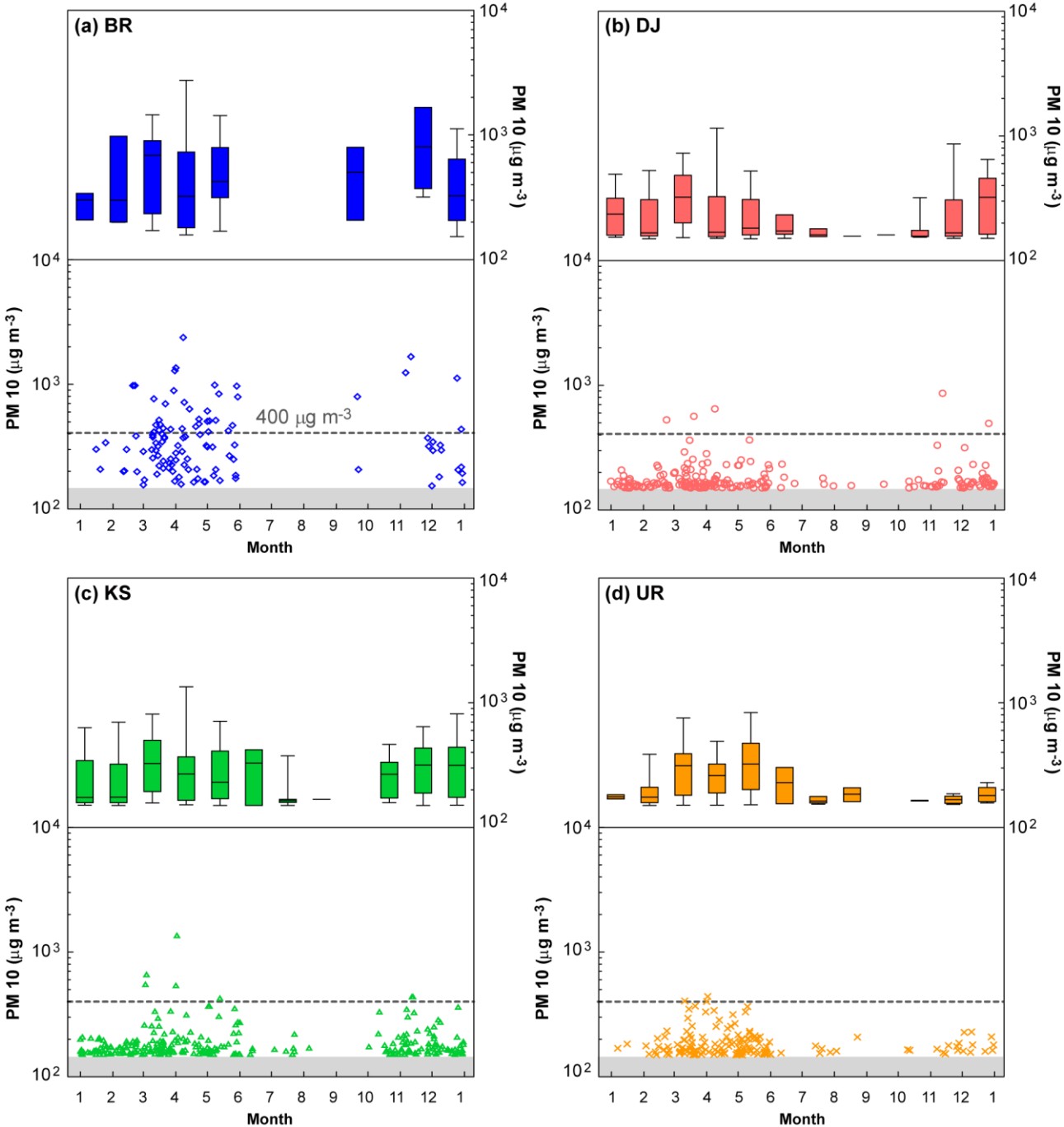

**Figure 5: Monthly variations of PM10 dust air concentration at (a) BR, (b) DJ, (c) KS, (d) UR. The lower panel shows a distribution of individual PM10 dust air concentration. Data symbols and reference lines are as in Fig. 4.**

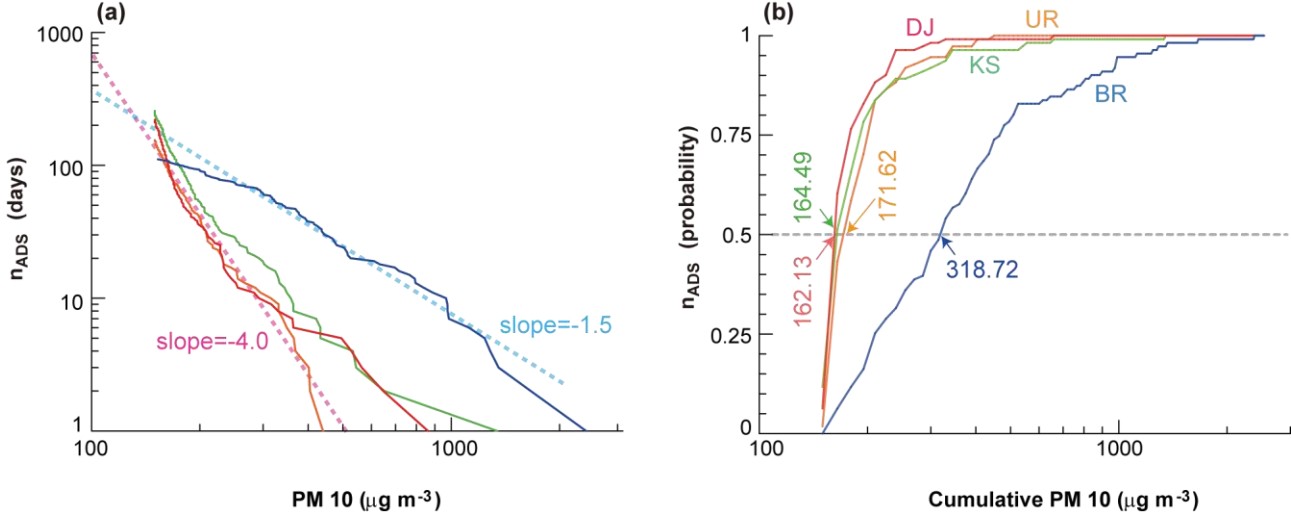

Figure 6: (a) Power-law fit to the ADS events as a function of PM10 dust air concentration. The two reference lines are the maximum and minimum asymptotic power fitting with slopes of -4.0 and -1.5, respectively. (b) Cumulative distribution of PM10 dust air concentration for the ADS data. Median values of PM10 dust air concentration (PM10 median) were 318.72 μg m$^{-3}$ for BR, 162.13 μg m$^{-3}$ for DJ, 164.49 μg m$^{-3}$ for KS, and 171.62 μg m$^{-3}$ for UR.

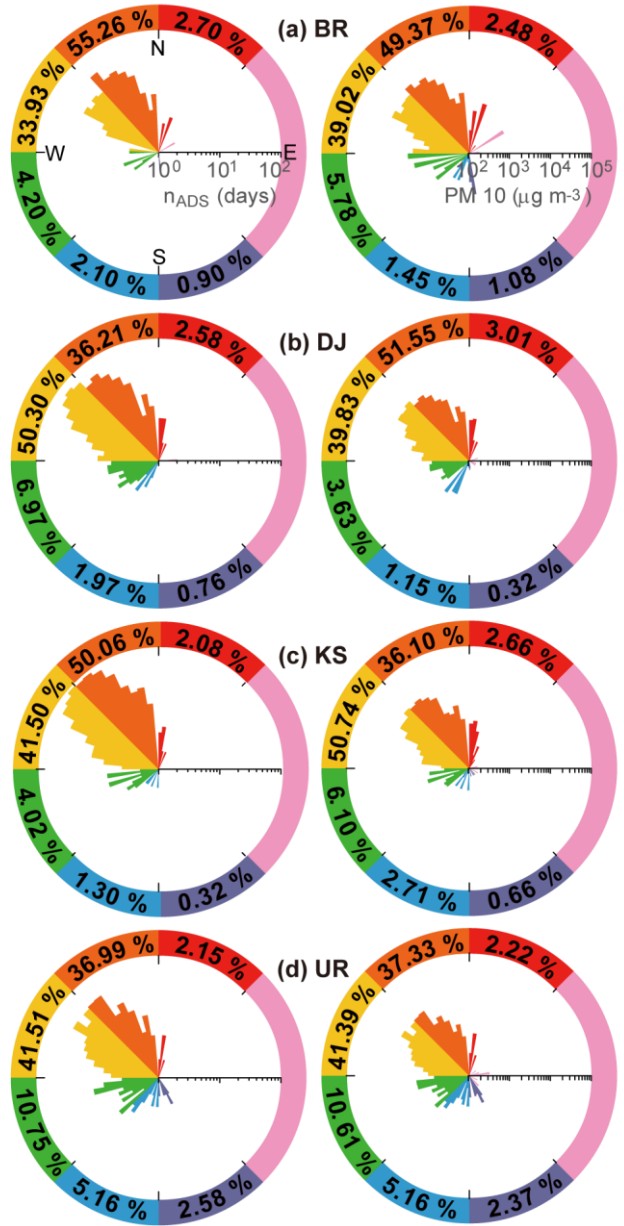

**Figure 7: Spatial distribution of ADS on a Rose diagram for (a) BR, (b) DJ, (c) KS, (d) UR. The left panel shows an angular distribution of number of days each year with ADS events ($n_{ADS}$). The right panel shows an angular distribution of PM10 dust air concentration.**

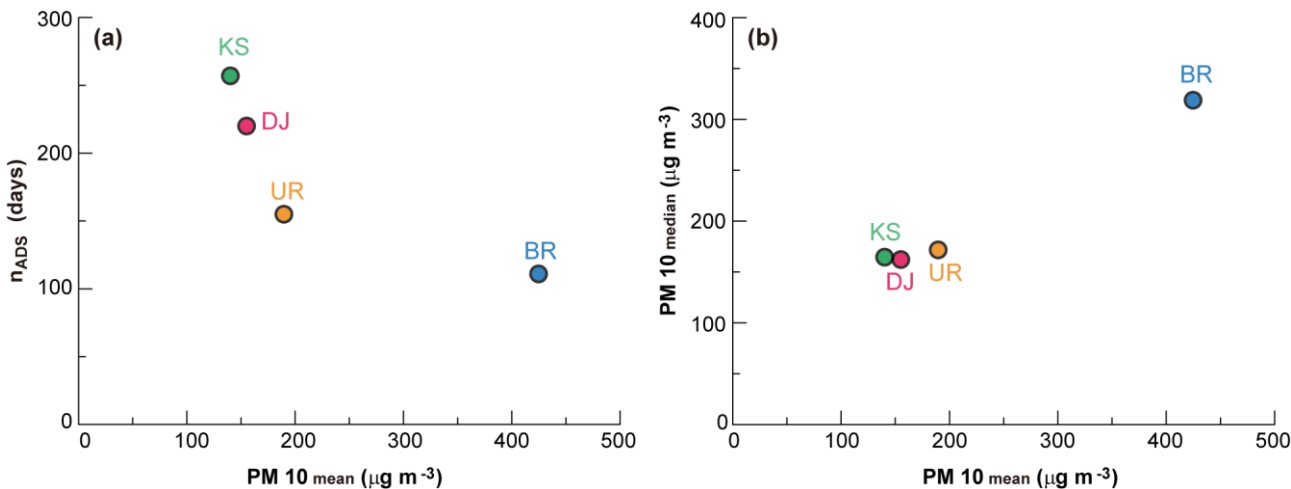

**Figure 8: (a) An inverse correlation between ADS occurrence ($n_{ADS}$) and annual mean PM10 dust air concentration (PM10 $_{mean}$). (b) A positive correlation between median PM10 dust air concentration (PM10 $_{median}$) and annual mean PM10 dust air concentration (PM10 $_{mean}$).**