# Peer review of "Identification of Atmospheric Transport and Dispersion of Asian Dust Storms"

_Natural Hazards and Earth System Sciences, 2016_

## Author Response (AR1)

List of Revision

Our response to referee's comments are listed as Red (Referee 1) and Blue (Referee 2) in a marked pdf file.

Blue (Referee 2)

(Referee 2's Comment 1) *This paper presents a short work on the identification of Asian dust storms outbreaks (ADS) affecting Korea. The period from January 2003 to August 2015 is analysed, and a total of 743 ADS affecting Korea are identified by means of Lagrangian trajectories. The HYSPLIT model is used to compute backtrajectories reaching the area of study at 1000, 1500 and 2000 m. This information is combined with observations of PM10 dust concentrations at 4 representative sites located in Korea. The main problem with the present version of the manuscript is that right now is a technical brief report. The analysis conducted are not scientifically sounding. There are some misunderstandings in the manuscript: the authors confuse the definition of concentration with density, presenting measurements of PM10 dust air concentration as dust densities.*

(Response to Referee 2's Comment 1)

It is true that we measured "PM10 dust air concentration" not the "dust density".

(Revisions according to Referee 2's Comment 1)

Throughout the text, we have revised manuscripts accordingly.

"dust density"           to           "PM10 dust air concentration"

"mean dust density"      to           "mean of PM10 (PM10 $_{mean}$)"

"median dust density"    to           "median of PM10 (PM10 $_{median}$)"

(Referee 2's Comment 2) *The lack of a Conclusion sections is a clear indication about the limitation of the analysis presented in this work.*

(Response to Referee 2's Comment 2)

We have added a new section for "Conclusion".

(Revisions according to Referee 2's Comment 2)

From Line 27 of Page 6 to Line 7 of Page 7

"The present study dealt with the Asian dust storms (ADS) outbreaks affecting Korea from January 2003 to August 2015. A total of 743 ADS air parcel backward trajectories reaching to Korea were identified by means of Lagrangian integrated trajectory (HYSPLIT) at three different ending altitudes at 1000, 1500, and 2000 m. In all four stations where ADS was monitored, we found that ADS occurrence rate was increased recently. Such increase of ADS occurrence was statistically significant in 99.9 % confidence level regardless of the threshold time divide of 1997/98 or 2000/01. Monthly variation of ADS occurrence was definitely non-uniform, as ADS was mostly concentrated in colder seasons of winters and springs. Instead, ADS events rarely occurred from June to September. Majorities of ADS events are azimuthally confined in narrow intervals of 290–340° on angle histograms, indicating that northwesterly distribution of dust transport was prominent. Such angular dependence of ADS occurrence agrees well with the higher PM10 dust air concentration from the northwest. We propose that the total amount of cumulative PM10 discharge was rather constant over time in Korea, as there is an inverse correlation between ADS occurrence and PM10 dust air concentration. Such constant PM10 flux allows weaker PM10 concentration for longer transport, and vice versa."

Red (Referee 1)

(Referee 1's Comment 1)

*The backward trajectory of each dust storm was calculated using HYSPLIT. But there is no description of HYSPLIT and its advantages and disadvantages, except for a short sentence on P. 2. Many readers of NHESS are probably not familiar with HYSPLIT, a free downloadable model from the NOAA website, and will appreciate having some background information on the model.*

(Response to Referee 1's Comment 1)

For a broader audience, a brief introduction to "HYSPLIT" is necessary.

We have added five new sentences.

(Revisions according to Referee 1's Comment 1)

Lines 21-31, Page 2

"In the present study, we trace the air parcel trajectories of ADS using the hybrid single particle Lagrangian integrated trajectory (HYSPLIT) model (Draxler and Hess, 1998). The HYSPLIT has evolved from the earliest model in 1982 (Draxler and Taylor, 1982) from modelling long-range air parcel trajectories into simulations of pollutant transportation, dispersion, and deposition over global scales. As an open source, the HYSPLIT is available on the Web through the ARL READY system (http://ready.arl.noaa.gov/HYSPLIT.php), operated by the National Oceanic and Atmospheric Administration (NOAA) Air Resource Laboratory (ARL). The HYSPLIT model requires the meteorological data and vertical movement of atmospheric circulation as input, and it displays the analysis of the simulation outputs (Stein et al., 2015). One great advantage of using the Lagrangian HYSPLIT is that both forward and backward trajectories are available with local or global airflow patterns to interpret the transport of pollutants. The HYSPLIT model is continuously evolving to cope with turbulent mixing process and to incorporate higher temporal frequency data available from the meteorological data (Stein et al., 2015)."

(Referee 1's Comment 2) *The study mentions that the trajectories of air transport at altitudes of 1000, 1500, and 2000 m were traced and shows the trajectories in Fig. 1b. But after Fig. 1b, there is no more mention of these different altitudes. Are they involved in subsequent analysis?*

(Response to Referee 1's Comment 2)

Air transport at three different altitudes were handled independently.

We have added three new sentences.

(Revisions according to Referee 1's Comment 2)

Lines 1-5, Page 6

"The HYSPLIT backward trajectories at different altitudes of 1000 m, 1500 m, and 2000 m were counted as individual path in the present study. They do not show a meaningful difference statistically, implying that atmospheric turbulent mixing was minimal. Such directional consistency for different altitudes of the HYSPLIT model might result from the relatively low and flat geographic conditions. For instance, both eastern China and western Korea are low in elevation, with bridging shallow Yellow Sea which extends 900 km in North-South directions and 700 km in East-West directions."

(Referee 1's Comment 3) *In the Introduction, the authors explain that ADS contain "surficial minerals of natural origin (e.g., weathered soils) as well as pollutants of anthropogenic origin such as black carbon, heavy metals, and sulfates." But in the Discussion, the authors seem to identify the desertification of the Gobi and Taklamakan deserts as the main cause of the recent increase of ADS. Can pollutants of anthropogenic origin, such as from coal burning and industrial plants, be another reason for the increase?*

(Response to Referee 1's Comment 3)

Influence of anthropogenic particulate matters is now included.

We have added three new sentences.

(Revisions according to Referee 1's Comment 3)

Lines 21-25, Page 5

"In addition to natural pedogenic enhancement, we cannot ignore the contribution of anthropogenic particulate matters supplied by fossil fuel combustion, coal burning and industrial plants. Although anthropogenic particulate matters represent only 5-30% of ADS volumetrically, they are harmful as they have a strong tendency to react with heavy metals preferentially. Considering on-going demand for the fossil-fuel combustion, it is reasonable to suggest that pollutants of anthropogenic origin are also responsible for the increase of ADS."

(Referee 1's Comment 4) *The Discussion concentrates on the results from the four stations in South Korea. Are there studies from other countries such as Japan and Taiwan to support the findings of the study on the recent increase and seasonality of ADS?*

(Response to Referee 1's Comment 4)

Seasonality of Asian Dust Storm in other neighboring countries are included in discussion.

We have added two new sentences.

(Revisions according to Referee 1's Comment 4)

Lines 6-8, Page 5

[revised manuscript text omitted]

---

## Author Response (AR2)

List of Revision

Our response to referee's comments are listed as Red (Referee 3) and Blue (Referee 4) in a marked pdf file.

Referee 3 had 3 major and 5 minor comments.

Referee 3's Major Comments

(Comment 1) *The lack of the HYSPLIT model description, scientific methods must be outlined clearly.*

(Response to Comment 1)

Physical principles of the HYSPLIT was extended as follows:

✓ Line 28, Page 2 – Line 2, Page 3

The spirit of Lagrangian HYSPLIT model relies on the determination of air concentration, as a cumulative summation of dust flow per unit grid cell. Each dust flow is considered as an independent particle flow puffed by advection, and is represented by its trajectory. Backward trajectories are constructed on the basis of Stochasitc Time Inverted Lagrangian Transport (STILT) model. The STILT is a widely used model for tracing atmospheric mixing between the source and the receptor point in terms of 2-dimensional upstream surficial fluxes.

(Comment 2) *Pollutant transport and dispersion are affected by atmospheric dynamics, fluid physical phenomena that occur in the atmosphere, and physical laws that govern them. These may facilitate or constrain transport and dispersion. All these topics are not considered in this manuscript.*

(Response to Comment 2)

Description on the pollutant transport and dispersion was explained as follows:

✓ Lines 8-11, Page 3

The HYSPLIT model describes transport and dispersion dynamics of aerosol, incorporating boundary stability determined by turbulent velocity, wind-blown dust emission algorithm, convectional plume rise produced by buoyancy of heat, wind velocity, atmospheric friction velocity, and in-cloud wet scavenging.

(Comment 3) *Pollutant transport and dispersion are affected also by different scales of motions as microscale, mesoscale, and synoptic scale. The authors make reference to this point only indirectly, as in section 3 in line 17 they commented that the data files to run HYSPLIT model comes from NCEP/NCA reanalysis, that I suppose with horizontal resolution of 2°, but they left many questions unanswered about this topic.*

Resolution of NCEP/NCA Reanalysis-1 was introduced.

✓ Line 32, Page 3 – Line 2, Page 4

The NCEP/NCAR Reanalysis-1 (https://www.esrl.noaa.gov/psd/data/gridded/data.ncep.reanalysis.html) provides meteorological data with grid resolution of 2.5° every 6 hours, regarding the vertical distribution of global aerosol and cloud from 1958 to present.

Referee 3's Minor Comments

(Comment 1) *In the introduction, in line 29, you must change irradiation by radiation, they are different concepts.*

(Response to Comment 1) Line 29, Page 1

Revised as "radiation".

(Comment 2) *Which version of the HYSPLIT model do you have used?*

(Response to Comment 2) Lines 5-6, Page 3

September 2015 version was now added.

(Comment 3) *You have used data from NCEP/NCAR analysis and you claim in the last sentence of the introduction that your results correspond to a local scale, are you sure?*

(Response to Comment 3) Line 25, Page 2

"in local scale" is specified as "in Eastern Asia"

(Comment 4) *Do you can explain the reason to consider the HYSPLIT backward trajectories at the altitudes of 1000 m, 1500 m and 2000 m, could you have considered backward trajectories at different heights?*

(Response to Comment 4) Lines 4-5, Page 4

Backward trajectories at cloud forming height was explained.

(Comment 5) *Please, use dust air concentration, not dust density.*

(Response to Comment 5)

Revised as suggested throughout the context.

Referee 4 had 5 minor comments.

(Comment 1) *The authors concluded a recent increase of ADS occurrence rate that was statistically significant in 99.9 % confidence limit, regardless of the locations of the observed stations and the threshold time divide. This is robust. But it seems that there may be also a decline in $n_{ADS}$, say after ~2007-08 (figure 2)? The trend seems clearly and consistently present at different stations. It is possible to test this signal statistically, similar as what has been done in Table 3 but with different year intervals. Some relevant discussion is necessary.*

(Response to Comment 1)

We tested a new temporal threshold of year 2007/2008, which was statistically insignificant. However, statistical significance was barely failed to discriminate the null hypothesis. Results were included in Table 3, and discussion was extended accordingly.

✓ Lines 21-24, Page 5

It is also apparent that there may be also a decline in ADS occurrence with respect to 2007/2008. Results for Welch's t–test are in the order of 10-2, slightly over the statistical threshold for the acceptance of the null hypothesis. Despite its failure of the null hypothesis, we cannot completely rule out the possibility as the temporal separation in 2007/2008 leaves only 8 data-points for recent intervals (Table 3).

(Comment 2) *Descriptions of model input data. In the Data section, the authors mentioned that they use meteorological data, including air pollution monitoring, such as in-situ dust density measurements. This is very general. What specific meteorological data were used as input parameters in the HYSPLIT modelling? What data were used directly for time-series analysis? More detailed and clear data description would be useful for the readers to understand the modelling and the results (Fig.2-8). Moreover, are there any criteria and reasons for choosing these four particular meteorological stations from 28? I guess it is possibly concerned with spatial distribution and longer coverage of observation periods etc.*

(Response to Comment 2)

As the referee pointed out, meteorological data collected from 4 stations were selected on the basis of efficient spatial and temporal coverage.

✓ Lines 20-23, Page 3

These stations are located at the western front (BR), southern edge (KS), eastern tail (UR), and central region (DJ) in South Korea, respectively. They were selected on the basis of spatial distribution and longer decadal coverage of observation periods.

The input data that we used were retrieved from the NCEP/NCAR Reanalysis-1 site at (https://www.esrl.noaa.gov/psd/data/gridded/data.ncep.reanalysis.html)

✓ Lines 27-32, Page 3

To trace the ADS provenance source, online version of backward trajectories HYSPLIT model (September 2015) was used. We used inputs of meteorological data from the National Centers for Environmental Prediction (NCEP; http://www.ncep.noaa.gov/) and the National Center for Atmospheric Research (NCAR: https://ncar.ucar.edu/) Reanalysis-1. Vertical motion of aerosol was adopted from the "Model Vertical Velocity" option on the HYSPLIT model (September 2015). Results of backward trajectories were displayed on ArcGIS program.

(Comment 3) *Please consider moving the added description of HYSPLIT model from the Introduction section to the second section (Data and analysis)?*

(Response to Comment 3) Section 2

A new section was made to include the extended explanation of HYSPLIT.

Section 2 The HYSPLIT Model

(Comment 4) *There are some repetitions of texts (for example, lines 22-23 at page 4 and lines 26-27 at page 3). Similar repetitions also occur for other texts, particularly in the result and discussion sections. Please check through.*

(Response to Comment 4)

Repeated sentences were deleted.

(Comment 5) *Page 5, line 4: "regional" may be a better word to replace "local" in the sentence "seasonal variation of ADS is a local, natural phenomenon in Eastern part of Asia".*

(Response to Comment 5) Line 30, Page 5

Revised as "regional".